# Synthesis of Nano-Praseodymium Oxide for Cataluminescence Sensing of Acetophenone in Exhaled Breath

**DOI:** 10.3390/molecules24234275

**Published:** 2019-11-23

**Authors:** Qian-Chun Zhang, Wu-Li Yan, Li Jiang, Yu-Guo Zheng, Jing-Xin Wang, Run-Kun Zhang

**Affiliations:** 1School of Biology and Chemistry, Key Laboratory of Chemical Synthesis and Environmental Pollution Control-Remediation Technology of Guizhou Province, Xingyi Normal University for Nationalities, Xingyi 562400, China; jiangli@xynun.edu.cn (L.J.); zhengyuguo@xynun.edu.cn (Y.-G.Z.); 2School of Public Health, Guangdong Pharmaceutical University, Guangzhou 510310, China; yanwuli@163.com (W.-L.Y.); wjx@gdpu.edu.cn (J.-X.W.)

**Keywords:** cataluminescence, sensor, acetophenone, Pr_6_O_11_

## Abstract

In this work, we successfully developed a novel and sensitive gas sensor for the determination of trace acetophenone based on its cataluminescence (CTL) emission on the surface of nano-praseodymium oxide (nano-Pr_6_O_11_). The effects of working conditions such as temperature, flow rate, and detecting wavelength on the CTL sensing were investigated in detail. Under the optimized conditions, the sensor exhibited linear response to the acetophenone in the range of 15–280 mg/m^3^ (2.8–52 ppm), with a correlation coefficient (*R*^2^) of 0.9968 and a limit of detection (S/N = 3) of 4 mg/m^3^ (0.7 ppm). The selectivity of the sensor was also investigated, no or weak response to other compounds, such as alcohols (methanol, ethanol, *n*-propanol, *iso*-propanol, *n*-butanol), aldehyde (formaldehyde and acetaldehyde), benzenes (toluene, *o*-xylene, *m*-xylene, *p*-xylene), *n*-pentane, ethyl acetate, ammonia, carbon monoxide, carbon dioxide. Finally, the present sensor was applied to the determination of acetophenone in human exhaled breath samples. The results showed that the sensor has promising application in clinical breath analysis.

## 1. Introduction

In recent years, the development of cataluminescence (CTL)-based gas sensors have drawn extensive interest, mainly because of the high sensitivity, fast response, long-term stability, and simplicity of the device [1,2,3]. CTL is defined as the light emission during the catalytic oxidation reaction on the surface of catalytic solid material [4,5]. This phenomenon was firstly reported by Breysse et al. in 1976 [6]. After that, Analytical chemists have made great efforts to promote the development of CTL. Nowadays, CCL was proved to be a powerful technology for rapid analysis, which shows promising applications in environmental analysis [7,8,9], food analysis [10,11,12], clinical breath analysis [2,13,14], and biochemical analysis [15,16].

Sensing material plays an important role in a sensor system, it interacts with analytes to induce detectable signals. The characteristic of the sensing material directly affects the sensitivity, selectivity and stability of the sensor. In 2002, Zhang et al. first introduced nanomaterials as sensing material into the design of CTL gas sensor, they demonstrated that nanomaterials can greatly enhance CTL performances, due to the pleasing advantages of high surface areas and high reaction activity for nanomaterials [17]. Since then, many nanomaterials were prepared and used for design of CCL sensors. For instance, nanomaterial Zn/SiO_2_-based sensor for ethyne [18], nanosized NaYF_4_:Er-based sensor for ketone [19], sensor based on *α*-MoO_3_ nanobelts for diethyl ether [20], sensor based on nano-MgO_2_ for *iso*-butanol [21], sensor based on nanomaterial g-C_3_N_4_ for carbon monoxide [22], nanomaterial NiO for H_2_S sensor [23], nanosized ZnO-based sensor for propionaldehyde [24].

There are abundant volatile organic compounds (VOCs) in human exhaled breath, some of them have been identified as biomarkers of diseases [25,26]. Analysis of concerned VOCs in exhaled breath has been recognized as a useful method for diagnosis of disease such as diabetes [27,28,29], asthma [30,31], lung cancer [32,33,34] and breast cancer [35,36]. The most outstanding advantage of breath analysis is that the pain-free and noninvasive collection of sample represents minimal risk to the human subjects [13,37]. Gas chromatography or chromatography/mass spectroscopy (GC/MS) are the traditional methods for breath analysis. However, these methods require complicated, bulky and expensive instruments, and their measuring procedures are also relatively tedious [38,39]. Sensors for breath analysis have attract wide attention because of their simple and compact device fabrication, as well as easy operation [40,41]. Examples include a WO_3_-based sensor for diagnosis of diseases [42], carbon nanotube (CNT)-based sensors functionalized with ionic liquids for infectious diseases [43], surface-enhanced Raman scattering sensor based on gold nanoparticles on reduced graphene oxide for gastric cancer [44]. 

Exhaled acetophenone in human breath is considered as an important biomarkers of breast cancers patients [45,46]. In the present work, nano-praseodymium oxide (nano-Pr_6_O_11_) was synthesized by hydrothermal method. The prepared nano-Pr_6_O_11_ exhibited strong CTL response to acetophenone, but no or weak responses to other compounds. Thus, the nano-Pr_6_O_11_ was used as sensing material to design CTL sensor for acetophenone. The sensor showed high sensitivity to acetophenone with a limit of detection of 4 mg/m^3^ (0.7 ppm). We demonstrated that the proposed sensor can be used for analysis of acetophenone in human exhaled breath. To the best of our knowledge, this work is first report of a CTL sensor for acetophenone. 

## 2. Results and Discussion 

### 2.1. Characterization

The physical-chemical characteristics such as morphology, size, and crystal form of nanomaterial directly affect the CTL behaviors. The surface morphology of the prepared Pr_6_O_11_ was investigated by SEM. As Figure 1a shows, quasi-hexagonal nanoparticles having about 50 nm diameters were obtained. Figure 1b shows the X-ray power diffraction patterns of Pr_6_O_11_. The diffraction peaks at 28.2°, 32.7°, 47.7°, 56.7°, 59.3°, 69.8°, 77.0°, 79.4° can be attributed to the (111), (200), (220), (311), (222), (400), (331), (420) planes. All of diffraction peaks are well indexed to a Pr_6_O_11_ (JCPDS.NO.41-1219) according to the standard card.

### 2.2. Selectivity of the CTL Sensor Based on Nano-Pr_6_O_11_

The selectivity of the CTL sensor based on nano-Pr_6_O_11_ was investigated. In a total of 18 kinds of chemical compounds include acetophenone, iso-propanol, benzophenone, n-propanol, n-butanol, methanol, ethanol, formaldehyde, acetaldehyde, n-pentane, ethyl acetate, toluene, o-xylene, m-xylene, p-xylene, ammonia, carbon monoxide, carbon dioxide, at concentration of 100 mg/m^3^ were tested. As shown in Figure 2, acetophenone produce strong CTL intensity, iso-propanol, n-propanol and *n*-butanol just induce weak response. The intensities of *iso*-propanol, benzophenone, n-propanol and *n*-butanol are 5.5%, 4.1%, 3.1% and 2.6% of the acetophenone. In addition, no detectable CTL signals were observed for other compounds. Under the same conditions, ceramic heater without coating with nano-Pr_6_O_11_ was used to detection of acetophenone, no CTL signal was detected, indicating that appropriate nanomaterial is essential to the CTL emission of acetophenone. 

It was reported that CTL responses are different for a given chemical compound on different nanomaterials, and the same nanomaterial shows dissimilar CTL responses to different chemical compounds. Although the detail reaction mechanism of CTL still remains uncertain, it is now well recognized that CTL emission must meet some requirements. One of the essential requirements is that the chemical compound can be oxidized to form a CTL intermediate, then the intermediate can produce photoemission when it returns to the ground state during the reaction process. It means that not all the chemical compounds can emit CTL on the same nanomaterial, even if they could be oxidized on the surface of nanomaterial. Although it is difficult for us to clarify the mechanism behind the reaction of acetophenone on nano-Pr_6_O_11_ surface, the above results demonstrate that the prepared nano-Pr_6_O_11_ exhibits satisfactory selectivity for sensing of acetophenone. 

### 2.3. Optimization of Working Temperature

During the CTL sensing process, working temperature has a great effect on the CTL response. The effect of working temperature on the CTL response was investigated by changing the working temperature from 199 to 278 °C. As shown in Figure 3, the CTL intensity increases with temperature from 199 to 258 °C and decreases when the temperature over 258 °C. This phenomenon may be attributed to the reaction rate increases with increasing working temperature, but higher working temperature accelerate molecular motion that may results in quenching of CTL intensity. In addition, higher working temperature emits stronger thermal radiation, leading to higher background noise is recorded. These factors determine the CTL signal and the S/N (signal-to-noise ratio) increase firstly then decrease with increment of working temperature. Because S/N reaches the maximum at 248 °C. Therefore, working temperature of 248 °C was chosen for further exploration.

### 2.4. Optimization of Air Flow Rate

The effect of the air flow rate on the CTL intensity was investigated by changing the flow rate in a rage of 180–250 mL/min. As shown in Figure 4, the CTL intensity increases gradually with the air flow rate from 180–230 mL/min, and reaches the maximum at 230 mL/min. Whereas, the CTL intensity becomes decline when the flow rate over 230 mL/min. This trend possibly results from the total reaction rate at low flow rate is controlled by the diffusion rate of acetophenone, and thereby CTL intensity increases with increasing flow rate. However, when the flow rate exceeds a certain degree, higher flow rate leads to insufficient reaction time, resulting in decrease in CTL intensity. Therefore, the optimal flow rate of 230 mL/min was selected for sensing of acetophenone.

### 2.5. Optimization of Detecting Wavelength

In order to investigate the effect of detecting wavelength on CTL response, a series of optical filters, including 380, 400, 425, 440, 460, 490, 535, 555 and 575 nm were selected in sequence to detect acetophenone using nano-Pr_6_O_11_ as sensing martial. The CTL emission from the catalytic oxidation of acetophenone on the surfaces of nano-Pr_6_O_11_ and the S/N values at different emission wavelength are shown in Figure 5. It can be seen that the CTL intensity gradually increases with the wavelength from 380–460 nm and then decreases when the wavelength over 460 nm. The maximum emission wavelength is observed at 460 nm. However, the background noise emitted by thermal radiation increases dramatically with the wavelength, and the maximum S/N is observed at 425 nm. As a result, 425 nm was selected as the detecting wavelength for sensing of acetophenone. 

### 2.6. CTL Response Profile and Analytical Characteristics

The CTL response profiles of acetophenone on the surface of nano-Pr_6_O_11_ were investigated by sensing of acetophenone at different concentrations under the above-optimized conditions. As shown in Figure 6a, the shape of the CTL response profiles is similar to each other. The CTL intensity sharply increases from the baseline to maximum value within 5 s after sample injection, the time of the CTL intensity decays from maximum value to baseline is about 20 s. These results demonstrate the capacities of rapid response and fast recovery of the sensor for acetophenone.

Under the optimized conditions, the CTL intensity is proportional to the concentration of acetophenone and exhibits a linear range of 15–280 mg/m^3^ (2.8–52 ppm), and a detection limit of 4 mg/m^3^ (0.7 ppm) is obtained at S/N = 3. The linear regression equation is *Y* = 100.9 *X* – 1356 (correlation coefficient *R*^2^ = 0.9968), where *Y* is the relative CTL intensity, *X* is the concentration of acetophenone (Figure 6b). Compared to the previously reported sensors for acetophenone [47,48], the present sensor shows wider linear range and lower LOD. The detail can be seen in Table 1.

### 2.7. Sample Analysis

Acetophenone in exhaled breath is identified as a biomarker of breast cancer. In order to probe the potential application of the designed sensor, 100 mL of exhaled breath samples from three volunteers were collected in sampling bags, then were detected by the CTL sensor based on nano-Pr_6_O_11_. However, no signal was measured, possibly due to the concentration of acetophenone was too low to be detectable. The three samples were spiked with different concentrations of acetophenone, subsequently, the spiked sample were measured by the sensor for recovery analysis, the results are shown in Table 2. The recoveries of acetophenone in the three samples are 106.7–113.1%, the RSDs are 3.2–4.0%. The satisfactory results demonstrate the potential of the sensor for real sample analysis.

## 3. Experimental Section

### 3.1. Materials and Instrumentation

Praseodymium nitrate hexahydrate, urea, acetophenone, *iso*-propanol, *n*-propanol, *n*-butanol, methanol, ethanol, formaldehyde, acetaldehyde, *n*-pentane, ethyl acetate, toluene, *o*-xylene, *m*-xylene, *p*-xylene, ammonia were purchased from Aladdin Reagent Co. Ltd. (Shanghai, China). Carbon monoxide and carbon dioxide were purchased from The National Standard of Material Resources Network (Beijing, China). 

The morphology of Pr_6_O_11_was characterized by using a scanning electron microscopy (Helios G4 CX). X-ray diffraction patterns were recorded on a Rigaku Ultima IV X-Ray Diffractometer using CuKα radiation with a scan speed of 5°/min ranging from 10° to 80°. (λ = 1.54 Å, operated on 40 mA and 40 kV current). BPCL ultra-weak luminescence analyzer (Guangzhou Microphotonics Technologies Co., Ltd., Guangzhou, China) was used to measured luminous signal. The luminescence analyzer equipped with a photomultiplier detector, a high voltage of 680 V was applied to the photomultiplier, the Data acquisition rate was set as 0.5 s.

### 3.2. Synthesis of Praseodymium Oxide Nanoparticles

1.0 g of praseodymium nitrate hexahydrate and 6.5 g of urea were dissolved in 240 mL deionized water under ultrasonically vibrating for 20 min. The resulting solution was dried at 100 °C for 2 h. Subsequently, the obtained gels were rinsed with deionized water and ethanol. Finally, the product was calcined at 500 °C for 4 h in air to obtain the target products.

### 3.3. Procedure for Sensing

The CTL sensor for acetophenone was fabricated according to our previous work [13,49]. In brief, 3.0 mL of deionized water was mixed with 0.5 g of solid catalyst, then the suspended liquid containing catalyst was dripped onto the surface of a ceramic heater and was heated to 450° for 20 min. The ceramic heater coating with catalyst was put into a lab-made quartz tube that with a gas inlet and outlet. A voltage regulator was used to control the temperature of the ceramic heater by adjusting the output voltage. An air pump was used to support air carrier, and the flow rare was controlled by a flowmeter. The CTL signal was measured and processed by the BPCL ultra-weak luminescence analyzer. The detecting wavelength was selected by choosing optical filter. At the beginning of each series of experiments, the CTL sensor based on praseodymium oxide was heated to 450° for 10 min in air to activate the catalyst, and eliminate the influence of previous absorbates. Then the temperature of the sensor was reduced to a certain value for subsequent detection.

## 4. Conclusions

In summary, we synthesized nano-Pr_6_O_11_ via a simple hydrothermal method, and demonstrated that the prepared nanomaterial can be used as sensing material for CTL detection of acetophenone. The CTL sensor exhibited good selectivity, fast response, and high sensitivity for sensing acetophenone. The sensor was applied to the determination of acetophenone in exhaled breath samples, satisfactory recoveries in the range of 106.7–113.1% were obtained. This work provides a simple, rapid, and sensitivity method for sensing of acetophenone in many cases, especially in exhaled breath.

## Figures and Tables

**Figure 1 molecules-24-04275-f001:**
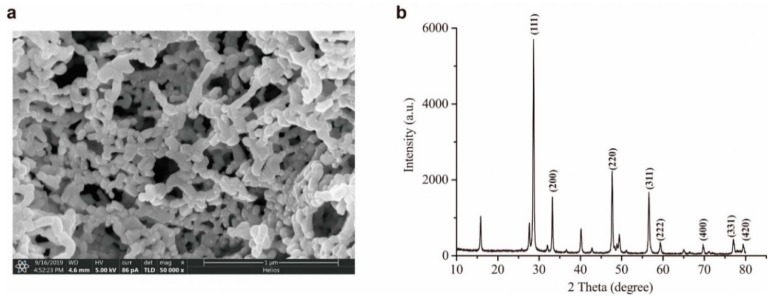
SEM images (**a**) and XRD pattern (**b**) of Pr_6_O_11_.

**Figure 2 molecules-24-04275-f002:**
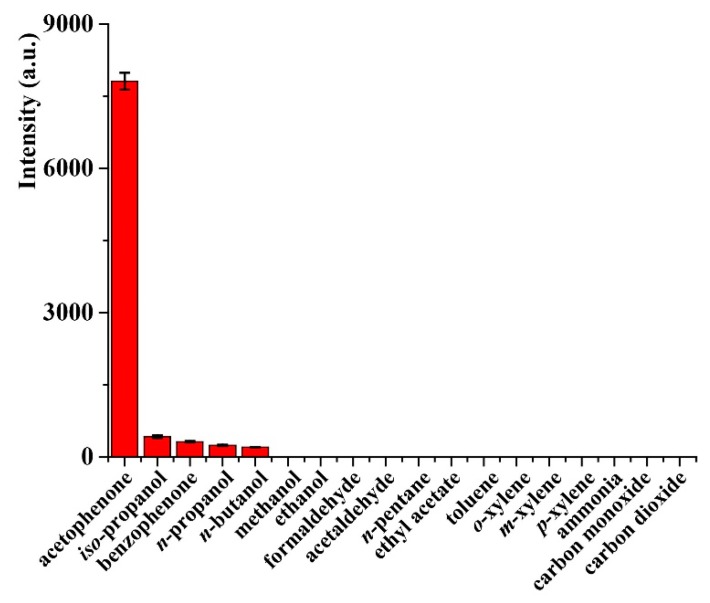
The response of sensor based on nano-Pr_6_O_11_ to different compounds. Temperature, 248 °C; air flow rate, 230 mL/min; detecting wavelength, 425 nm, concentration, 100 mg/m^3^. Error bars stand for ±S.D (standard deviation).

**Figure 3 molecules-24-04275-f003:**
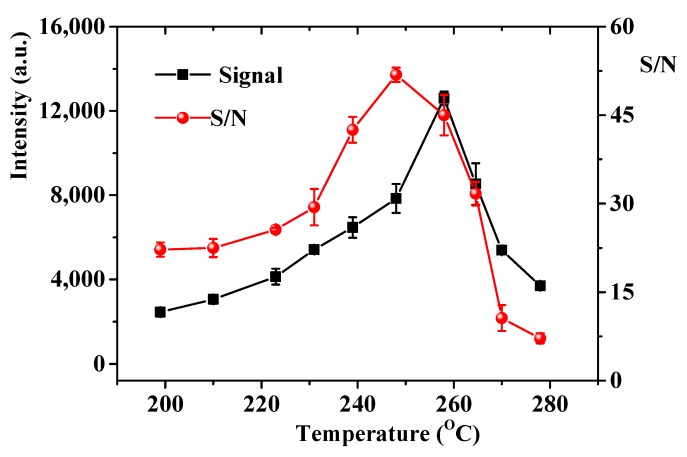
Effect of temperature on the cataluminescence (CTL) intensity and the S/N. Air flow rate, 230 mL/min; detecting wavelength, 425 nm, concentration, 100 mg/m^3^. Error bars stand for ±S.D.

**Figure 4 molecules-24-04275-f004:**
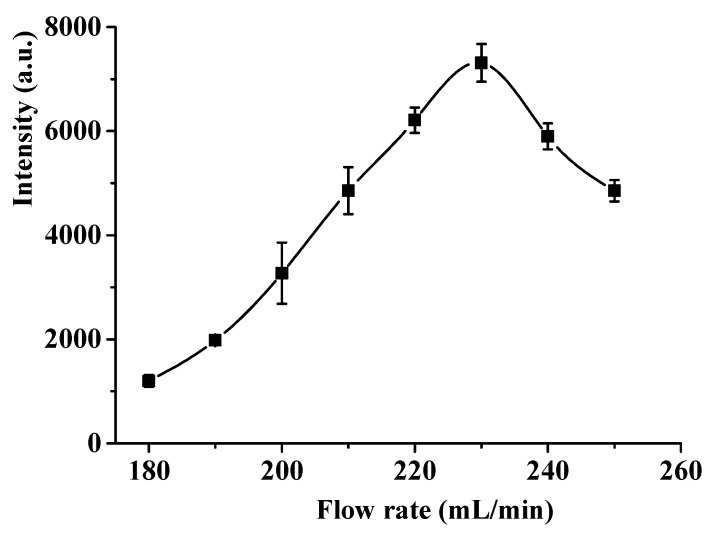
Effect of air flow rate on the CTL intensity. Temperature, 248 °C, detecting wavelength: 425 nm, concentration, 100 mg/m^3^. Error bars stand for ±S.D.

**Figure 5 molecules-24-04275-f005:**
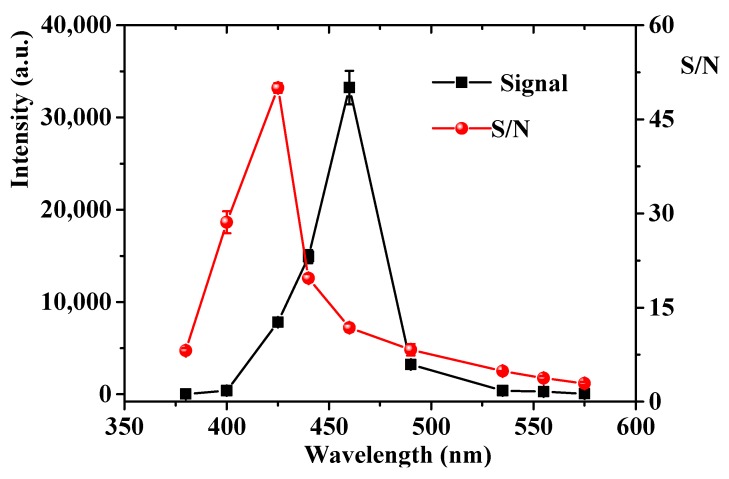
CTL spectra emission on nano-praseodymium oxide. Temperature, 248 °C; air flow rate, 230 mL/min, concentration, 100 mg/m^3^. Error bars stand for ±S.D.

**Figure 6 molecules-24-04275-f006:**
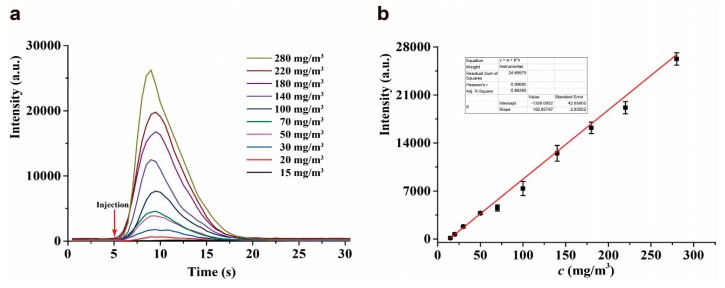
(**a**) The CTL response profiles. (**b**) The calibration curve for acetophenone. Temperature, 248 °C; air flow rate, 230 mL/min; wavelength, 425 nm. Error bars stand for ±S.D.

**Table 1 molecules-24-04275-t001:** Comparison of the analytical characteristics of the sensors for acetophenone.

Principle	Sensing Materials	Linear Range (ppm)	LOD (ppm)	References
CTL	Nano-Pr_6_O_11_	2.8–50	0.7	Present work
Electrochemistry	1-Octyl, 3-methylimidazolium tetrafluoroborate	5–80	2.0	[47]
Quartz microbalance	Macrocyclic oligolactams	5–40	2.0	[48]

**Table 2 molecules-24-04275-t002:** Determination of acetophenone in the exhaled breath samples spiked with acetophenone by the designed sensor (*n* = 3).

Sample No.	Spiked Concentration (mg/m^3^)	Measured Concentration (mg/m^3^)	Recovery (%)	RSD (%)
1	20.0	21.9 ± 0.9	109.5	4.0
2	25.0	26.7 ± 0.9	106.7	3.2
3	30.0	33.9 ± 1.3	113.1	3.8

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
