# Peer review of "Synthesis of Nano-Praseodymium Oxide for Cataluminescence Sensing of Acetophenone in Exhaled Breath"

_molecules, 2019, doi:10.3390/molecules24234275_

Round 1

Reviewer 1 Report

This manuscript describes the synthesis and use of nano-Pr6O11 particles and their function in the selective sensing of acetophenone through the detection of catalytic luminescence.  The manuscript is clear and at first glance describes the physical and sensor properties of the system.  However, there are major scientific questions that must be addressed before publication.

Figure 1 needs a size bar that is readable.  Figure 2 needs more description of the excitation and emission wavelengths for the compounds tested.  Is 425 nm the excitation or emission?  Is there a spectrum to look at? Figure 6 A is confusing because the data look like they are collected as a function of time, but start at 5 seconds?  Perhaps this is a delayed fluorescence signal?  Is there control data of the foreign agents without nanoparticles?

There are minor grammatical edits needed.  Line 21,38 and 41 are fragmented.  Two independent clauses need "and, or but" and the comma.  Otherwise, they can be separated by a period as two standalone sentences. There are similar errors of this kind throughout the document.

 A description of the chemical interaction that makes these particular nano oxide material selective to only acetophenone, compared to the other foreign substances of similar chemical structure, would make this manuscript more compelling.  Perhaps the nanoparticles are aggregated or perturbed in some way gives way to the CTL.  Alternatively, n-pi* conversion to triplet acetophenone could be one reason why these systems produce luminescence (or phosphorescence). It's possible that benzophenone would also produce a luminescence signal, but that foreign agent was not tested.  Otherwise, all the foreign agents tested are primarily alcohol based, or are other chemical structures with different functional groups.  The phosphorescence and related photophysical processes are sensitive to temperature, especially if they occur through a thermally activated process or produce a delayed photoluminescence signal.  Therefore, it's not clear whether the nanoparticles have some effect on the CTL process.  The authors need to review the literature and explain.

Here are a few manuscripts describing these interesting photophysics from a brief literature search:

Talanta. 1986 Jan;33(1):17-25. J. Chem. Phys. 56, 3044 (1972); https://doi.org/10.1063/1.1677640 https://www.sciencedirect.com/science/article/abs/pii/003991408680008X

Reviewer 2 Report

In this manuscript, Q.-C. Zhang and his co-workers present the fabrication and characterisation of the first praseodymium oxide nanomaterial-based cataluminescence acetophenone sensor, and the performance testing of this sensor including selectivity, sensitivity, linearity range, detection limit, and working-conditions (working temperature, air-flow rate, detecting wavelength, CTL response profile). The work is carefully done; therefore, it deserves publication. To improve the manuscript, however, authors should consider the following comments:

--- Authors should provide a brief explanation of the temperature dependence of CTL response and signal/noise (chapter 2.3.).

--- Authors should provide a brief explanation of the CTL intensity decrease upon increasing the airflow rate above 230 mL/min (chapter 2.4.).

--- Similarly, the signal/noise dependence on the detecting wavelength requires a brief explanation.

--- Authors have to provide information about the fabrication of the acetophenone sensor, at least briefly, by naming the major steps. Simply writing that “The CTL sensor for acetophenone was fabricated according to our previous work [13, 47].” is not enough. Both a brief explanation and references are required.

--- A comparison of the performance of the fabricated CTL acetophenone sensor with other acetophenone gas sensors published in the literature should be provided at the end of the manuscript e.g. in a table, including at least LOD, linearity range, and sensor platform.

--- A final check is required for improving the English of the text. (e.g. include vs. including; “In addition, there is no detectable CTL signals was observed” vs. “In addition, no detectable CTL signal was observed….; etc..”

Round 2

Reviewer 1 Report

Just a final grammatical and sentence structure check is required (line 129 and 130)  Otherwise, the authors have addressed the comments and the manuscript is recommended for publication.

Author Response

We tried our best to improve the manuscript and made some changes in the manuscript. And here we did not list the changes but marked in red in revised paper.

We appreciate for Reviewers’ warm work earnestly, and hope that the correction will meet with approval.
Once again, thank you very much for your comments and suggestions.